# Finite Element Simulation and Experimental Investigation of Nanostructuring Burnishing AISI 52100 Steel Using an Inclined Flat Cylindrical Tool

**Victor Kuznetsov** [1,*], **Igor Smolin** [2] , **Andrey Skorobogatov** [1] **and Ayan Akhmetov** [2]

1 Department of Heat Treatment and Physics of Metal, Ural Federal University, 620002 Yekaterinburg, Russia
2 Institute of Strength Physics and Materials Science SB RAS, 634055 Tomsk, Russia; smolin@ispms.ru (I.S.); akhmetov.a.zh@ispms.ru (A.A.)
* Correspondence: wpkuzn@mail.ru

**Abstract:** This article is devoted to the development of a sliding burnishing scheme using a flat cylindrical indenter. The previously established patterns of nanostructured state formation in the AISI 52100 steel subsurface layer showed a need to create a special tool with a variable tilt angle of the indenter and with force regulation. A new tool with a cubic boron nitride indenter opens wide possibilities for nanostructuring burnishing of hardened bearing steel. Firstly, a flat cylindrical indenter has high durability due to repeated rotation around its axis. Secondly, the change of the tilt angle to the treated surface allows controlling the contact compression pressure and plastic shear deformation, which determines the formation of a nanostructured state of the material by the method of severe plastic deformation (SPD). The purpose of the work is to determine the optimal parameters of the process and tool in order to form a nanostructure and significantly increase surface layer microhardness. The goal was achieved by the methods of finite element modeling (FEM) and experimental studies of burnishing when the indenter tilt angle changes from 0.5° to 2.5° under dry processing conditions. Numerical simulation of the process made it possible to establish optimal values of the indenter tilt angle of 2° and the burnishing force 250 N according to the criteria of maximum contact pressure and cumulative deformation. The experimental studies of cumulative deformations and the coefficient of friction by the method of burnishing a split disc and dynamometry of the process confirmed the FEM results. The transmission microscopy, durometry, and 3D surface profilometry showed the sensitivity of nanocrystallite sizes, microhardness, and roughness to an indenter tilt angle and confirmed the optimality of the established tilt angle value.

**Keywords:** nanostructuring burnishing; cylindrical indenter; FEM; temperature field; stress and strain; constitutive relations; friction coefficient; microhardness; surface topography; transmission microscopy

## 1. Introduction

Burnishing is a method of smoothing and strengthening machine parts by surface plastic deformation. The method is implemented in modern machining centers, where a moving tool is pressed into the surface of a rotating workpiece. Traditionally, the indenter is made of diamond and has a hemispherical shape, but in recent years, other materials and forms of the indenter have been proposed, along with kinematic schemes for processing cylindrical and flat surfaces with the application of advanced methods of surface layer material nanocrystallization.

In [1], a cylindrical polycrystalline diamond tool was used to burnish the LY12 aluminum alloy. The burnishing mechanism was investigated on the basis of the laws of contact mechanics. The simulation model brings into focus the interrelationship of the burnishing force, the tool geometry, the parameters of the technological process, and the mechanical properties of the material.

Korzynski et al.'s work [2] presents the burnishing of the shaft surface using a tool with a cylindrical indenter, the axis of which is perpendicular to the axis of the shaft being processed. The influence of burnishing parameters on the surface stereometric structure, surface layer strengthening, and residual stress distribution is investigated. It was found that the microhardness of the 42CrMo4 alloy steel surface layer after burnishing increased by 29%, and the compression stresses in the surface layer increased up to 400 MPa.

Studying the burnishing of 20Cr4 steel by a spherical indenter with a radius of 2 mm, Kuznetsov et al. [3] focused on the formation of a nanocrystalline structure in the burnished surface layers. In the previous works of Valiev et al. [4,5], it has been shown that the formation of a nanocrystalline structure in structural steels is achieved by intense plastic shear deformation under hydrostatic compression with a pressure of more than 2 GPa and a value of cumulative shear strain of more than two. Similar data are reported in [3]. Based on the results of studies given in this paper, it is possible to calculate the wedge angle of the plastically displaced metal to the plane of the processing surface, depending on the pile-up height, the indenter insertion depth, and the contact zone length. With a burnishing force of 200 and 340 N, a feed of 0.01 and 0.04 mm/rev, and a friction coefficient of 0.13 and 0.34, this angle varied in the range from $1°6'$ to $2°36'$.

The refinement of the structure of the quenched AISI 420 steel by nanostructuring burnishing with a spherical indenter made of polycrystalline synthetic diamond was studied in [6]. It is shown that at a burnishing force of 340 N, a feed of 0.04 mm/rev, and with a change in sliding speed from 6 up to 26 m/min, the plastic shear deformation ranges from 2.1 to 3.4. At the same time, a homogeneous nanocrystalline structure with grain sizes less than 100 nm is formed in the near-surface layer.

As an interesting illustration, we can take a work in which an indenter made of cubic boron nitride has a cylindrical shape and is inclined at an angle to the processing surface [7]. This work describes experimental studies of the nanostructuring of the 1.3505 (AISI 52100) quenched steel surface layer by a sliding cylindrical indenter inclined to a flat surface being treated at an angle of 2 degrees. The studies revealed the influence of the burnishing force, applied to the tool and varying in the range from 100 to 340 N, on the change in the thickness and microhardness of the strengthened layer. It was established for the first time that with nanostructuring surface strengthening with a normal force of 250 N, the microhardness in the surface layer with a thickness of up to 20 μm reaches more than 1000 $HV_{0.05}$. It is obvious that this method of burnishing with different inclination angles of the indenter needs further investigation with the application of various methods, including numerical analysis and experiment.

Thus, the literature review in the field of experimental studies of the burnishing process indicates that most of the works are devoted to using a traditional spherical shape of a diamond processing tool (85% according to the review [8]). In the case of a cylindrical tool shape, it is usually a long cylinder, the axis of which is parallel to the surface to be processed. Steels are the most commonly used material (64% according to the survey [8]), which is subjected to sliding burnishing. Friction conditions are not reported in more than $^1/_3$ articles, and only 9% of studies are devoted to dry friction conditions [8]. Therefore, the use of a disc-shaped tool made of cubic boron nitride with an axis tilt to the treated surface is innovative and little studied.

The numerical simulation of this mechanical finishing process has aroused growing interest in research and industry. Recently, a comprehensive and critical review of literature focusing on the modern advances in ball burnishing simulation using the Finite Element Method (FEM) numerical technique has been performed in [9]. This review involves up-to-date 2D and 3D numerical simulations of the ball burnishing treatment to study surface roughness, residual stresses, microhardness, plastic strain, phase transformation, elastic recovery effect, ball indentation mark, and thermal behavior.

In the work of J.T. Maximov et al. [10], finite element simulation of 37Cr4 steel diamond burnishing in the Abaqus package was used to predict the effect of process parameters

(speed, force, feed, and friction coefficient) on the change of residual stresses and temperature in the treated surface layer depending on the burnishing speed.

In the article [11], using the FEM method, J.T. Maximov et al. investigated the influence of the process parameters and the number of tool passes on the distribution of residual stresses and of equivalent plastic deformation. Based on the results, the fatigue endurance dependence on stresses and the number of cycles of 2024-T3 aluminum alloy loading were constructed.

The study of compression stresses and axial residual stresses during the diamond burnishing of EN 1.4301 steel with a spherical indenter $R = 3.5$ mm by the 2D FEM method was carried out in the work of C. Felhő and G. Varga [12]. The simulation takes into account the friction coefficient and the initial surface roughness after preliminary turning.

The effect of the friction force in the process of nanostructuring burnishing with a spherical indenter on the change of deformations and stresses in the contact zone by the dynamic FEM method was studied in [13]. The relationship between the burnishing force and the friction coefficient in the contact zone of the indenter and 20Cr4 steel with the height of the plastically displaced metal pile-up, the depth of the tool insertion, and the intensity of cumulative shear deformations is established. The regularities of changes in shear stresses along the surface layer depth in the contact and non-contact zones of the deformation focus are revealed.

Summing up the review in the field of numerical modeling of the burnishing process and related phenomena, it should be concluded that such studies are much less common than experimental ones (12% and 75%, respectively, according to [8]), the studies with a combined approach (numerical–experimental) are also rare. The main object of modeling is residual stresses and roughness. The processed object is more often modeled with an elastic-plastic material, and thermomechanical problem formulations are much less common (about 4% [9]). ABAQUS software is the most common means of modeling processes such as burnishing (38% according to the review [9]). Both 3D and 2D calculations are widely represented in the research works. Therefore, a combined study including experimental and numerical methods is in demand. There are no computer modeling studies in the literature considering the influence of burnishing process parameters and tool geometry on the conditions of nanostructured state formation in the processed steel surface layers.

The modern studies of the burnishing process with a cylindrical indenter are mainly devoted to the finishing treatment of the surfaces of workpieces, which are bodies of revolution. The development of the method of nanostructuring burnishing of flat parts with a cylindrical indenter located at an angle to the treated surface, proposed by the authors of this study in [7], is of current interest. The inclination of the cylindrical tool creates a wedge of the deformable material shear favorable for nanostructuring between the processing plane and the line connecting the indenter depth point and the point corresponding to the maximum height of the pile-up. It can be noted that the inclination angles of the cylindrical indenter and the wedge of the deformable material practically coincide. According to the research results [3], the wedge angle of the quenched steel being sheared varies from 1 to 3 degrees.

Of considerable interest are the studies on the application of an indenter made of CBN500 ceramic material for nanostructuring burnishing of AISI 52100 quenched roller-bearing steel.

Based on the literature review and the purpose of the work, the following research objectives and the sequence of their solution are formulated:

- Development of numerical and finite element simulations of the burnishing process with the application of a cylindrical indenter with an adjustable inclination angle to the treated surface;
- Dynamometric studies of the process and establishment of the relationship of contact forces and the friction coefficient with the inclination angle of the indenter;
- Simulation and experimental study of the influence of the cylindrical indenter inclination angle and of burnishing force on the contact pressure and shear deformation;

- Establishment of microstructure change regularities, microhardness, and roughness depending on the inclination angle of the cylindrical indenter and its burnishing force.

Since the value of the coefficient of friction of the CBN ceramic indenter in dry contact with AISI 52100 steel has not been studied before, for numerical simulation, it is necessary to initially conduct dynamometric studies of the process and establish the real values of the coefficient of friction at different tilt angles of the indenter.

## 2. Materials and Methods

### 2.1. Material for Treatment and the Tool

The material under processing is AISI 52100 steel. Chemical composition of steel (%): C—0.99; Si—0.3; Mn—0.45; Cr—1.49; Ni—0.3; S—0.021; P—0.027; Cu—0.25; Fe—the rest.

The samples under study were made in the form of disks with a diameter of 90mm and a thickness of 12 mm and were quenched at 830 °C with cooling and water tempering at 300 °C, after which their hardness was 58–60 HRC.

The tool (Figure 1) provides the possibility of changing the indenter inclination by an angle α to the reference plane of the treated surface. The cylindrical indenter 1 is installed on surface 2 of replaceable bushings 3, made with tilt angles of 0.5, 1.0, 1.5, 2.0, and 2.5 degrees and fixed with clamp 4. The burnishing force from 100 to 250 N is set by a spring and adjusted by a screw 5. By reinstalling screw 6, the angle of indenter rotation can be adjusted.

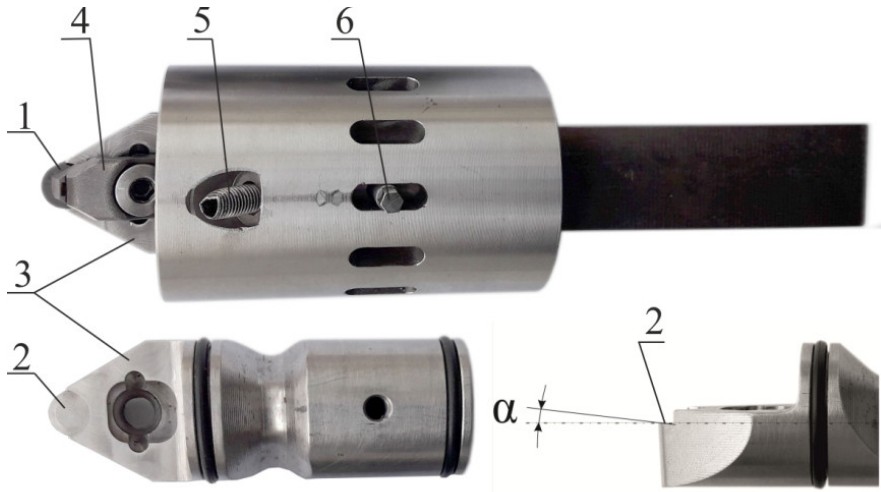

**Figure 1.** The tool with a variable angle of cylindrical indenter inclination to the treated surface and with burnishing force adjustment.

A SECO plate of the RNMN 090300 CBN010 type with a diameter of 9 and a thickness of 3.2 mm with a chamfer of 0.2 × 20° was used as a cylindrical indenter of the tool. The tool material of the plate, CBN010 cubic boron nitride, provides high strength and wear resistance during the finishing of bearing steels.

### 2.2. Mathematical and Finite Element Models

During the process of burnishing treatment, heat is produced due to plastic deformation of the material surface layer and friction between the tool and the workpiece surface. Since the strain and stress fields depend on the temperature field, and the change in stresses and plastic strains, in turn, affects the temperature field, the problem must be solved within the framework of coupled problems of thermomechanics [14,15].

In this case, simultaneous analysis of the temperature field due to heat generation and conduction as well as stress and strain fields is carried out. Local instantaneous temperatures can be rather high, so all material parameters (elastic moduli, heat capacity,

thermal conductivity coefficient, linear thermal expansion coefficient) are supposed to depend on temperature.

For a more detailed description of stress and strain distributions in the contact zone, we adopt a two-dimensional formulation of the problem in plane strain conditions because three-dimensional calculations with comparable accuracy are computationally expensive.

The central issue determining the level of stresses and deformations in the material under treatment is the choice of the constitutive relations and parameters of the selected model. To describe the deformation of steel, we adopt an isotropic elastoplastic model of the material. To define plastic strain rates, the $J_2$ plastic flow theory is adopted, which demands determining the flow stress. One of the most common models of metal material mechanical processing is the Johnson–Cook model [16]. This model allows us to take into account not only strain hardening but also the dependence on the strain rate and thermal softening at a temperature increase. In the Johnson–Cook model, the equation for the yield stress has the following form:

$$\sigma_y = (A + B\varepsilon^n)\left(1 + C \ln \dot{\varepsilon}^*\right)(1 - T^{*m}),$$ (1)

where $\varepsilon$ is the equivalent plastic deformation; $\dot{\varepsilon}^* = \dot{\varepsilon}/\dot{\varepsilon}_0$ is the normalized plastic deformation rate; $\dot{\varepsilon}_0$ is the specified rate of deformation at which the model parameters are determined (in the original article by Johnson and Cook and many other works, it is assumed that $\dot{\varepsilon}_0 = 1 \ \mathrm{s}^{-1}$); $T^* = (T - T_r)/(T_m - T_r)$ is homological temperature; $T_r$ is the specified temperature at which the model parameters are determined (usually a room temperature); $T_m$ is the melting point.

Thus, five material constants $A$, $B$, $n$, $C$, $m$, and some more parameters $T_r$ and $T_m$ are introduced. The values of the Johnson–Cook model parameters for AISI 52100 steel are presented in Table 1. There are parameter values from other authors' articles that are accepted in this study. The parameters of the model differ significantly from different authors, as well as the stress–strain diagrams of AISI 52100 steel, which can be seen, for example, in Figure 6 in the article [17].

**Table 1.** Johnson–Cook model parameters.

| Authors, Paper | $A$, MPa | $B$, MPa | $n$ | $C$ | $m$ | $T_r$, °C | $T_m$, °C |
|---|---|---|---|---|---|---|---|
| Poulachon et al. [18] | 11.032 | 4783 | 0.0946 | 0 | 1 | 0 | 775 |
| Guo et al. [19] and Ramesh et al. [20] | 688.17 | 150.82 | 0.3362 | 0.04279 | 2.7786 | | 1370 |
| Shrot et al. [21] | 635.926 | 101.703 | 0.649 | | 2.259 | 635.926 | |
| Guo et al. [22] and Bapat et al. [23] | 2482.24 | 1498.5 | 0.19 | 0.027 | 0.66 | | |
| This paper | 322 | 994 | 0.34 | 0.043 | 0.597 | 20 | 6510 |

Our parameters were determined by approximating our experimental data at different temperatures, which will be shown further on in the description of the experimental results. The $T_m$ parameter was considered not as the melting point, but as a model parameter that allows us to more accurately describe experimental data using Formula (1).

The temperature value $T$ to be used in Equation (1) is calculated from the heat balance equation (thermal conductivity). Two factors can be taken into account as heat sources. First, during deformation, a part η of the work of stresses on plastic deformations is converted into heat (it is usually assumed that the parameter η = 0.9). The heat flux per unit volume is computed using the equation:

$$\dot{Q}_{pl} = \eta \sigma \dot{\varepsilon}^{pl},$$ (2)

where $\sigma$ is the stress, $\dot{\varepsilon}^{pl}$ is the rate of plastic straining.

The second source of heat is the work of friction (due to frictional sliding). The rate of frictional energy dissipation is given by

$$\dot{Q}_{fr} = \tau \dot{\gamma}, \tag{3}$$

where $\tau$ is the frictional stress and $\dot{\gamma}$ is the slip rate. Some part of this energy releases as heat fluxes into the surfaces being in contact.

Since the simulation was proposed to be performed in the formulation of a plane strain field, there arose a problem of choosing the values of the corresponding burnishing forces. It is necessary to establish the relationship between the forces that are applied to the instrument in a real three-dimensional formulation and the specified forces in two-dimensional calculations of the plane strain field. We will carry out calculations in the SI unit system. This means that the width of the sample and the indenter in the contact zone will be equal to 1 m.

The basis for determining the correspondence between real forces and calculated ones is the condition that the compression stresses under the indenter in the two-dimensional case and in the real three-dimensional case are the same, $\sigma_{2D} = \sigma_{3D}$. Then the relationship between the forces in the two–dimensional case $P_{2D}$ and in the three-dimensional one $P_{3D}$ will be as follows

$$P_{2D} = P_{3D} \frac{S_{2D}}{S_{3D}}, \tag{4}$$

where $S_{2D}$ and $S_{3D}$ are the areas of the contact zones in the two–dimensional and three-dimensional cases, respectively.

To derive a further formula, it is important to determine which three-dimensional case we are considering. Firstly, our indenter is cylindrical. Secondly, we will assume that we exert pressure with this indenter on the plane surface of the sample.

Here are expressions for the areas of the contact zones in the two-dimensional and three-dimensional cases:

$$S_{2D} = l_c \cdot 1\,\text{m}^2, \; S_{3D} = \frac{k}{2} l_c \cdot L_c, \tag{5}$$

where $k > 1$, because the shape of the contact zone is of teardrop shape (parabolic), and not triangular, as assumed in Formula (5);

$L_c$ is the width of the contact zone in meters in the three-dimensional case, and the corresponding width of the indenter contact (perpendicular to the plane of simulation) with the workpiece in the case of two-dimensional simulation;

$l_c$ is the length of the contact zone in meters, and the corresponding contact length of the indenter with the contacting bodies' surface plane in the case of two-dimensional simulation in the formulation of a plane strain condition.

The diagram of the contact interaction of an inclined cylindrical indenter at an angle $\alpha$ with the workpiece is shown in Figure 2.

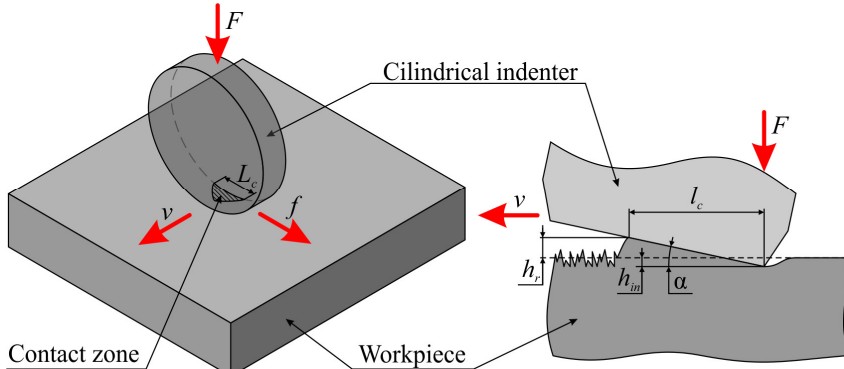

**Figure 2.** Contact interaction diagram of a cylindrical indenter inclined to a plane surface during nanostructuring burnishing.

With the insertion depth $h_{in} = 5$ μm and the diameter of the cylindrical indenter $d = 9$ mm, it is possible to estimate the width of the contact zone along the direction perpendicular to the plane of simulation as follows

$$L_c = 2\sqrt{h_{in} \cdot d - h_{in}^2} \approx \sqrt{h_{in} \cdot d} \approx 0.424 \text{ mm} = 4.24 \cdot 10^{-4} \text{ m.} \tag{6}$$

This estimate is from above (maximum), since it is assumed that the cylinder contacts the entire depressed surface. In fact, due to the elasticity of the real metal material, contact does not occur on the entire surface.

If we calculate $S_{3D}$ more precisely through the circular segment area and the corresponding area of the cone plane section, then we can get the value for the coefficient $k = 1.334$. Thus, to match the forces, we have the values given in Table 2.

**Table 2.** Correspondence of forces in three-dimensional and two-dimensional cases.

| $P_{3D}$, N | $P_{2D}$, kN |
|---|---|
| 100 | 354 |
| 150 | 531 |
| 200 | 708 |
| 250 | 885 |

The values of density, elastic modules, and other thermophysical parameters as well as their dependences on the temperature used in the calculations were taken from [24,25] and are presented in Appendix A.

The developed mathematical model formed the basis of the FEM model. The ABAQUS computer code was used for the numerical calculation of stress, strain, and temperature fields. The simulation was performed in two steps. In the first step, the quasi-static task of indenter penetration into the sample surface under the action of a constant burnishing force was analyzed using ABAQUS/Standard. Using the distribution of stresses and deformations from the first step, the subsequent analysis of the indenter movement with a sliding speed and under the action of the burnishing force was carried out in the second step. To solve a quasi-static problem, the ABAQUS/Explicit code was used. As known, the use of ABAQUS/Explicit (explicit solver) allows one to quickly perform quasi-static simulations using explicit dynamics in the case of complex constitutive relations and contact conditions if certain approaches are used [26,27].

The mesh density was determined by the applied loads and boundary conditions. Since a burnishing process leads to high deformations, stresses, and temperatures in the near-surface zone, a very fine mesh is required precisely at the contact zones of the indenter with the material being processed. Away from this zone, stresses and temperature gradients become low, small-scale, and a coarser mesh can be used there.

In this two-dimensional study, the workpiece had the shape of a rectangle with a length of 22 mm and a thickness of 9 mm. The FEM mesh consisted of 60,200 linear quadrilateral elements of CPE4RT type (4-node plane strain thermally coupled bilinear displacement and temperature). The grid contained 1721 nodes in length and 35 nodes in thickness (Figure 3). The smallest element of the mesh was square with a size of 10 μm. These dimensions were chosen based on a preliminary study on the mesh convergence of the resulting solution.

The indenter was considered a rigid body with its shape defined by an analytical surface in the form of a set of linear segments. To account for the heat sink in the indenter, its point heat capacitance was set in the calculations.

To simulate the contact with Coulomb friction between the indenter and the treated surface, a 'Surface to surface' type contact was adopted.

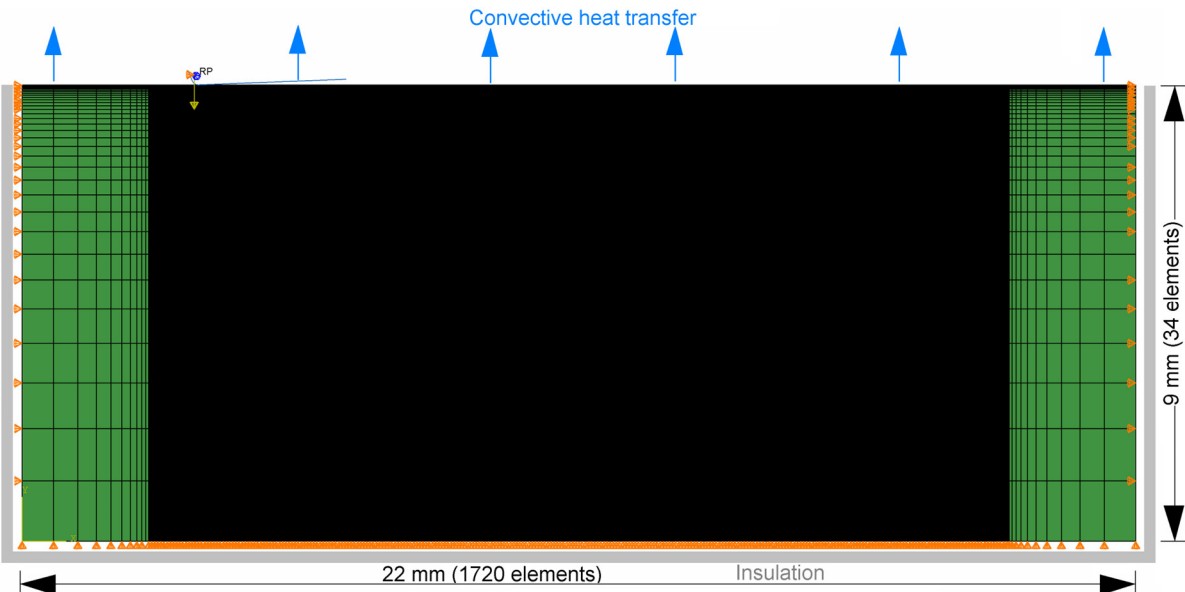

**Figure 3.** Schematic representation of FEM mesh with the boundary conditions.

## 2.3. Methods of Experimental Research

To implement finite element simulation of the process, experimental studies were carried out to determine a friction coefficient in the contact zone and the degree of plastic shear deformation in relation to an inclination angle of the cylindrical tool. The value of plastic shear deformation during nanostructuring burnishing was determined by the method of a split disk and by measuring the amount of material influx in one of its halves in depth from the surface.

The split disk consists of two halves (Figure 4). In each half, there are holes for dowel pins and a screw for their reliable fixation. The mating surfaces of the halves were lapped and polished to ensure uniform pressing over the entire surface.

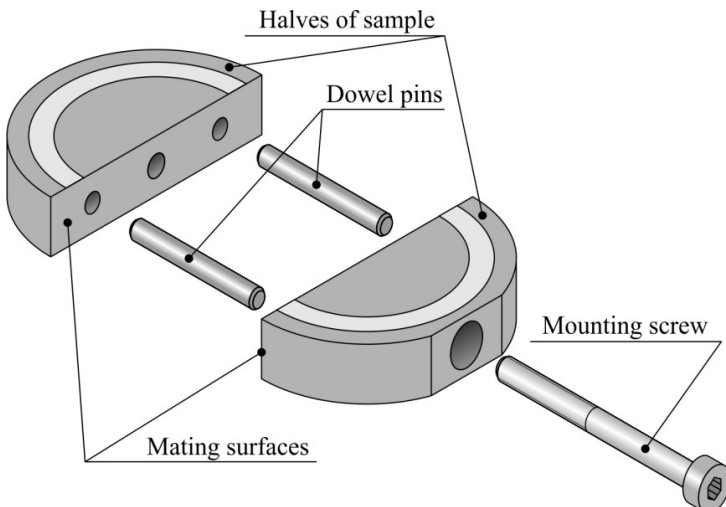

**Figure 4.** Split disk design.

In the process of nanostructuring burnishing (Figure 5a), the surface layer material undergoes shear, which at the junction of the disk halves leads to the formation of an influx on one half and a dint on the other. Analyzing the geometry of the influx or dint, it is possible to determine the distribution of shear deformation in the surface layer depth. The

degree of cumulative shear deformation $\varepsilon_{xy}$ at a distance $x$ from the burnishing surface is determined by the inclination angle $\gamma$ tangent to the influx boundary (Figure 5b).

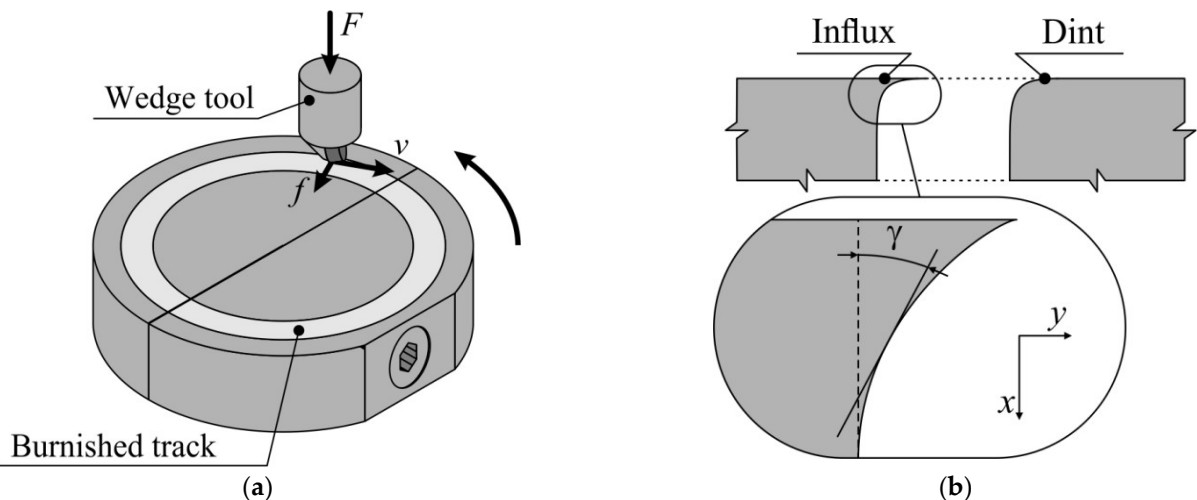

(**a**)  (**b**)

**Figure 5.** Experimental estimate of shear deformation in a split disc: (**a**) diagram of the tool movement when burnishing the track on the disc surface; (**b**) diagram to estimate shear deformation by the surface layer depth.

The value of cumulative relative deformation in the depth of the surface layer can also be determined by the formula

$$\varepsilon_{xy} = \mathrm{tg}\gamma = \frac{dy}{dx}. \tag{7}$$

The true deformation is determined by the logarithmic dependence

$$e = \ln(1 + \varepsilon_{xy}). \tag{8}$$

The diagram of contact forces, friction forces, and technological parameters of the process when treating a plane surface with a cylindrical indenter is presented in Figure 6.

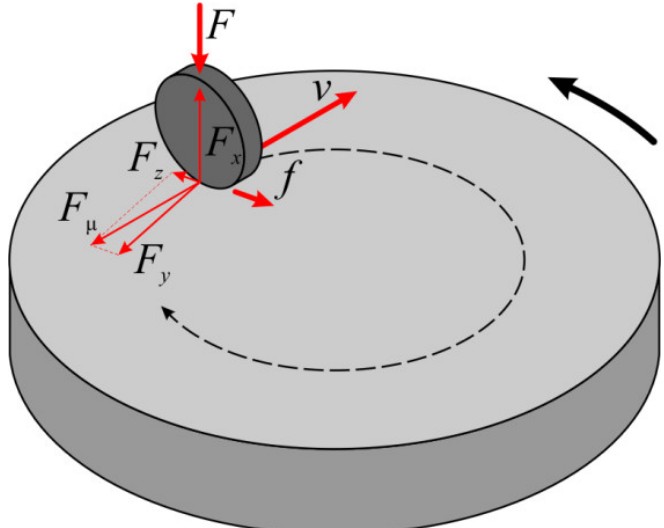

**Figure 6.** Diagram for determining contact and friction forces when processing a plane surface with a cylindrical indenter.

The coefficient of friction in the contact of the tool with the treated surface is determined by the formula

$$\mu = \sqrt{\frac{F_y^2 + F_z^2}{F_x^2}} \tag{9}$$

where $F_x$, $F_y$, $F_z$ are the contact forces acting in the direction of the coordinate axes.

Previously, the surfaces of the split disc were processed by finishing turning on a KNUTH V-TURN 410 (KNUTH Machine Tools, Germany) machine with a WMNG 080408 plate from Sandvik (SANDVIK Coromant, Sweden) at a speed of 80 m/min with a feed of 0.08 mm/rev. The cutting depth was 0.2 mm. The arithmetic mean deviation of the *Ra* profile after turning was measured by a Wyko NT-1100 3D (Veeco, USA) profiler and was 0.419 μm.

Further, during a single set-up, nanostructuring burnishing of three-ring tracks with a width of 7 mm on each side of the disk was carried out by an innovative tool. The tool was installed in the Kistler 9257BA (Kistler, Switzerland) dynamometer (Figure 7).

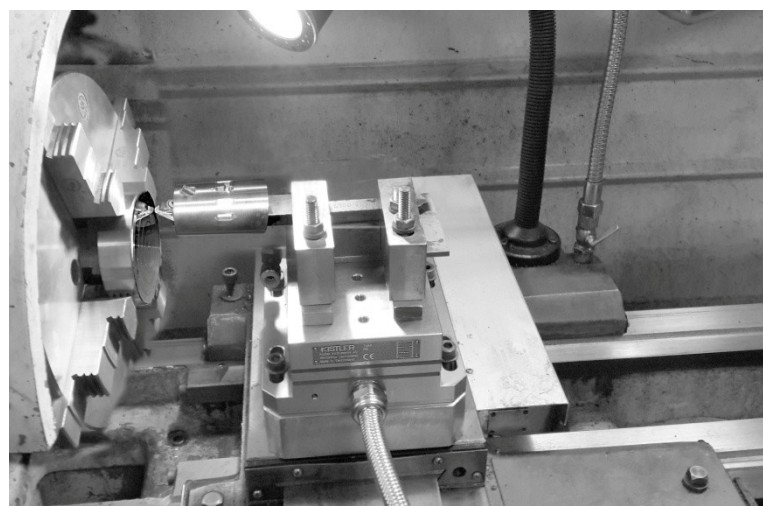

**Figure 7.** Nanostructuring burnishing of the split disc surfaces with a tool installed in a Kistler 9257BA dynamometer.

The study of shear deformation and contact forces was carried out with a burnishing force of $F$ = 250 N, feed of $f$ = 0.04 mm/rev, and burnishing speed of $v$ = 25 m/min under dry friction conditions. The choice of the burnishing force equal to 250 N is due to the results obtained in [7]. It is shown that with a burnishing force of 200–250 N, the thickness of the nanostructured layer formed by burnishing with a cylindrical indenter at an inclination angle of 2 degrees is 4.2–5.7 μm, while a decrease in the normal force to 100–150 N leads to a decrease in the modified layer to 1.2–1.9 μm, respectively.

Transmission microscopy was performed to confirm the formation of the nanostructured state in the AISI 52100 steel surface layer after nanostructuring burnishing. The microstructure was studied using a JEOL JEM 2100 (JEOL, Japan) transmission microscope. To prepare foils on the AgieCut Spirit 20 (AGIE, Switzerland) electroerosion machine from the experimental samples, the fragments of square section with a side of 5 mm and 12 mm thick were cut out according to the diagram shown in Figure 8.

Foil workpieces 0.3 mm thick were separated from the cut samples from the side of the burnished surface on a circular sawing machine using a 0.5 mm thick diamond wheel with continuous water cooling of the cutting zone. The foil workpieces were mechanically thinned on the opposite side to the burnished one and then smoothed on the polishing paper with a grain size of P200 (ISO-6344) [28] until a thickness of 100 μm was reached. After mechanical treatment, electrochemical polishing of the surface was carried out in

orthophosphoric acid. To prevent the burnished surface from interacting with acid during electrochemical polishing, the surface was covered with a thin Teflon film.

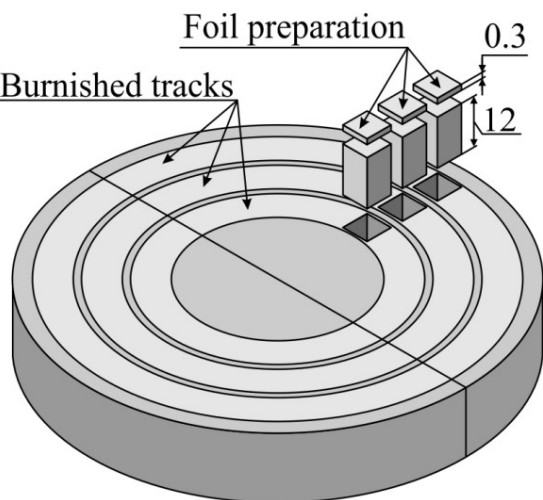

**Figure 8.** Diagram of cutting out samples to prepare foils.

### 3. Results

*3.1. Experimental Study of the Friction Coefficient and Cumulative Shear Deformation*

The results of dynamometric studies of dry nanostructuring burnishing show a significant influence of the indenter tilt angle on the magnitude of the reaction forces in contact in directions opposite to the sliding velocity ($F_y$) and feed ($F_z$) (Figure 9).

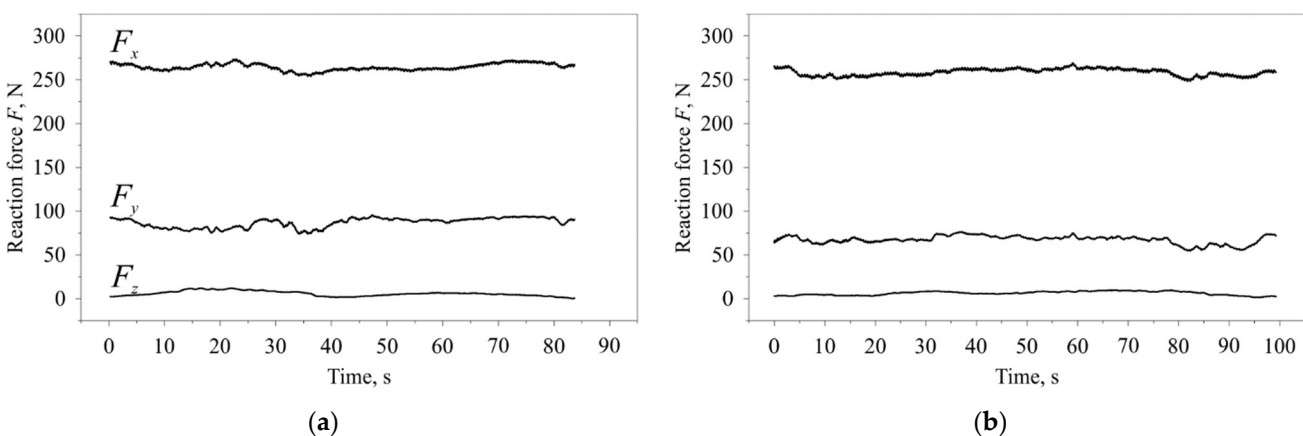

**Figure 9.** Dynamometry of reaction forces in contact with a cylindrical indenter: (**a**) Tilt angle of 0.5°; (**b**) Tilt angle of 2.0°.

The largest value of the forces $F_y$ and $F_z$ is observed at a small tilt angle of 0.5°, which causes the maximum friction coefficient $\mu = 0.33$, calculated by Formula (9) (Figure 10). As the tilt angle increases, there is a gradual decrease in the friction coefficient, which reaches a minimum of $\mu = 0.26$ at an angle of 2°. It is also important to note that during the entire track treatment, there are no trends in changes in contact forces, which characterizes the nanostructuring burnishing process as stationary.

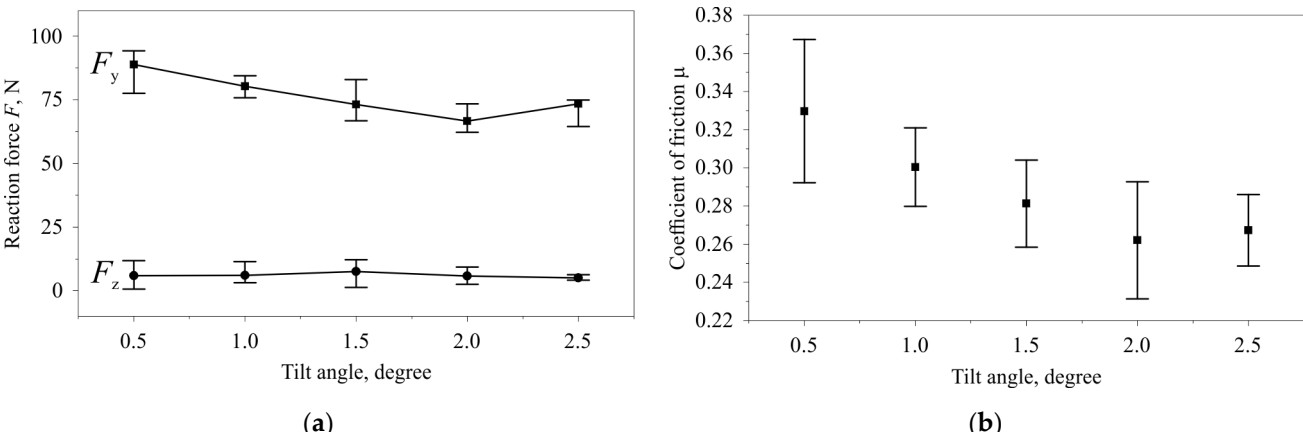

**Figure 10.** Dependence of the reaction forces (**a**) and friction coefficient (**b**) on the tilt angle of the indenter.

To determine the dependence of the shear component of the cumulative plastic deformation of the material after nanostructuring burnishing, the mating planes of the split samples near the treated surface are analyzed. The geometric parameters of the influxes were studied with the WYKO NT-1100 (Veeco, USA) optical 3D profilometer in PCI mode in several sections. The obtained results show a significant influence of the wedge angle on the amount of material influx (Figure 11).

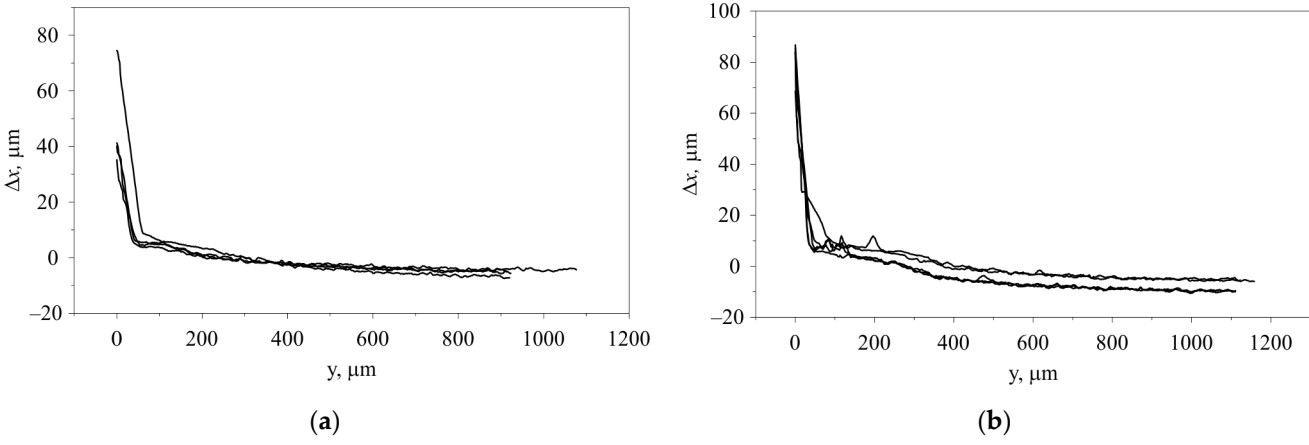

**Figure 11.** Profilometry of the material influx of the split sample: (**a**) tilt angle of 0.5°; (**b**) tilt angle of 2.0°.

The characteristic of the relative and true deformation determined on the basis of the measured geometric parameters of the material influx shows that the minimum value of deformation $e$ = 1.05 is observed at the indenter tilt angle of 0.5° (Figure 12). As the angle increases, there is a significant increase in true deformation reaching a value of 2.4 at an angle of 2°. A further increase in the angle to 2.5° leads to a decrease in true deformation to a value of 2.1.

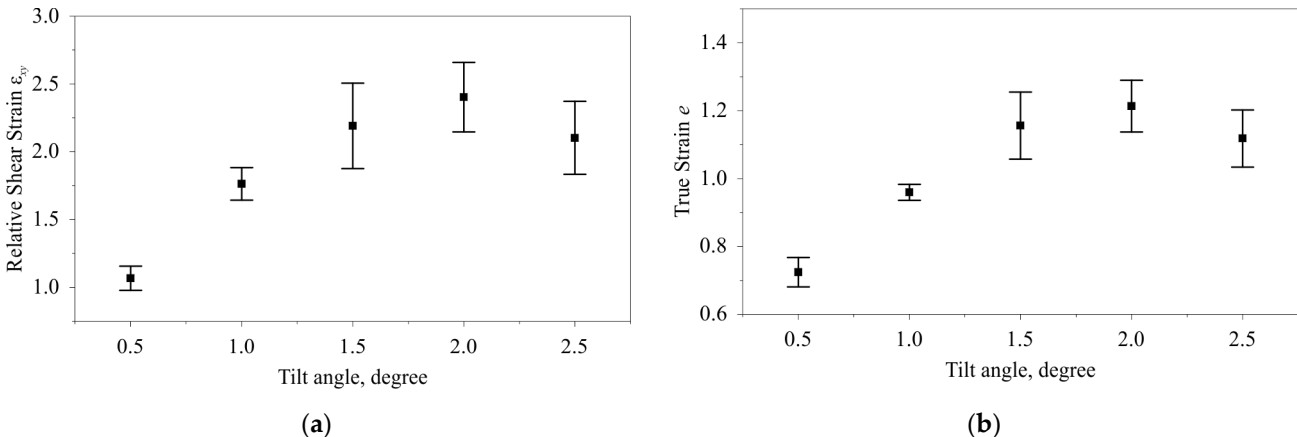

**Figure 12.** Dependence of the relative (**a**) and true (**b**) shear strain on the indenter tilt angle.

### 3.2. Numerical Results

With the help of the prepared geometric and finite element models, the process of burnishing of the AISI 52100 steel surface layers with an inclined cylindrical indenter was simulated at different values of burnishing force and indenter tilt angle, taking into account experimentally established values of the friction coefficient μ.

Besides, it was revealed how the values of the parameters of the selected constitutive relation (parameters of the Johnson–Cook model) and consideration of heat generation in the friction zone affect the results obtained.

As an example of the solutions obtained, Figure 13 shows the distribution of hydrostatic (or contact) pressure, with an indenter tilt of 2.5 degrees, a burnishing force of 250 N, and a friction coefficient of μ = 0.26.

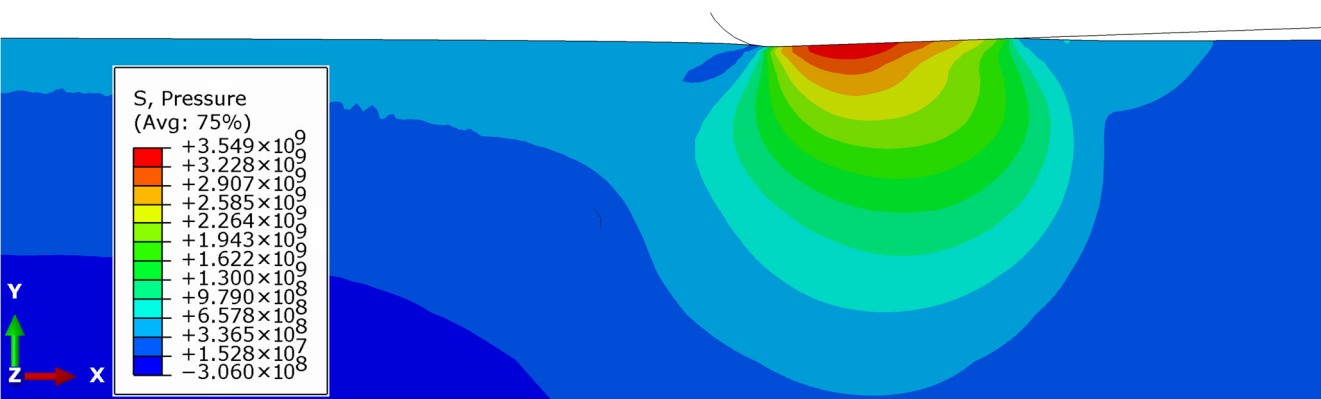

**Figure 13.** The distribution of hydrostatic pressure (Pa).

To study the influence of the tilt angle of the cylindrical indenter on the characteristics of the stress–strain state in the deformation focus, as the latter, we consider the maximum values of pressure, cumulative inelastic deformation, and temperature.

The dependence of the maximum pressure in the deformation focus on the burnishing force at different values of the indenter tilt angle is shown in Figure 14a. As the burnishing force increases, the pressure increases. It can be seen that at small angles of inclination, the dependences have a maximum in the range of 200–230 N. The dependences of the maximum cumulative plastic deformation in the deformation focus on the burnishing force are shown in Figure 14b. An increase in the burnishing force causes an almost linear increase in the cumulative deformation.

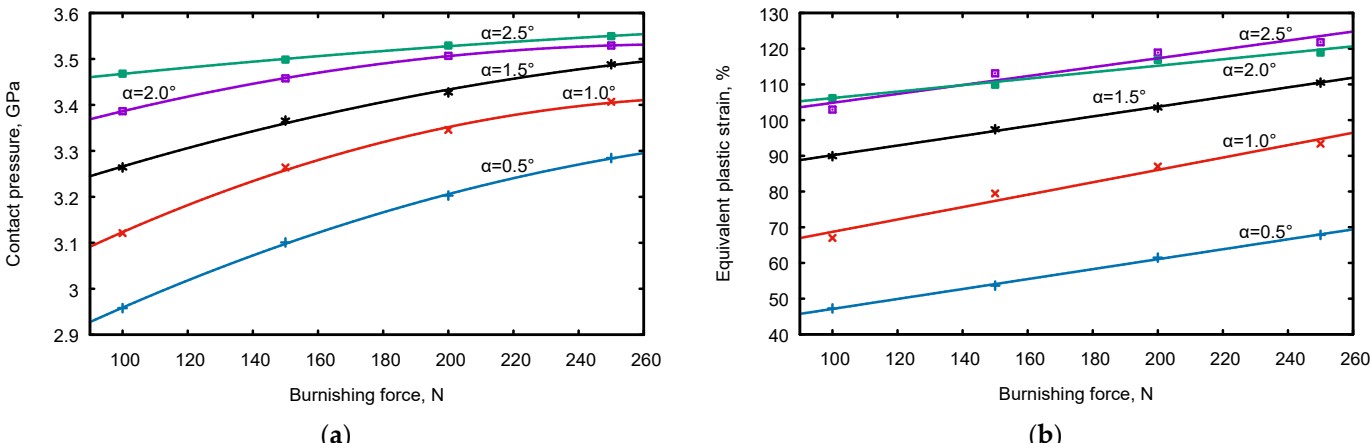

**Figure 14.** The dependences of (**a**) the contact pressure and (**b**) cumulative plastic strain on the burnishing force.

The dependences of the contact pressure and the maximum cumulative plastic deformation in the deformation focus on the tilt angle of the cylindrical indenter at different values of the burnishing force are shown in Figure 15. All the graphs have a maximum in the range of about 2 degrees. The analysis of the results shows that with an increase in the tilt angle, the contact pressure increases, which is due to a decrease in the length of the contact zone. At the same time, the gradient of change in the characteristics of the stress state in the deformation focus increases with a decrease in the indenter tilt angle.

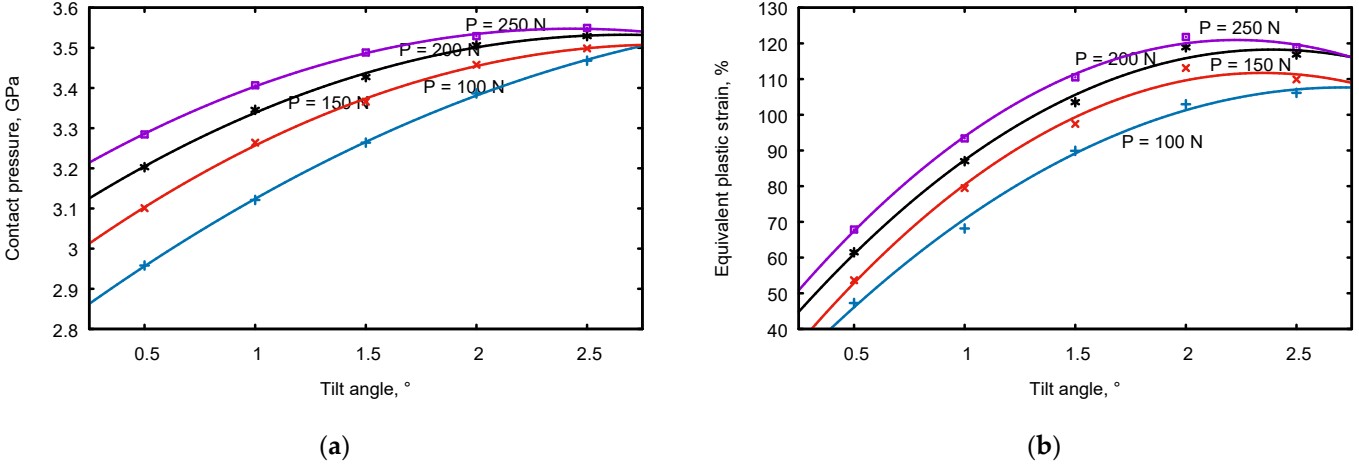

**Figure 15.** The dependences of (**a**) the contact pressure and (**b**) cumulative plastic strain on the tilt angle.

The important characteristics, from the point of view of providing conditions for nanostructuring burnishing, are the change in the sign of shear stresses and the absence of tensile horizontal stresses. It should be noted that there is no pronounced influence of the tilt angle of the indenter on these parameters. A change in the sign of shear stresses is noted for a tilt angle of 0.5°. With an increase in the burnishing force, the contact pressure increases slightly, which is explained by the fact that the length of the contact zone increases more noticeably. It should also be noted that there is no pronounced influence of the burnishing force on the level of shear stresses, and there is no change in the sign of shear stresses in the focus.

The calculations also revealed that taking into account heat generation during contact friction leads to a significant (about 4 times) increase in temperature in the deformation zone, which indicates the importance of its consideration. The temperature distribution

near the contact zone for such a calculation with a burnishing force of 250 N and an indenter tilt angle of 2° is shown in Figure 16.

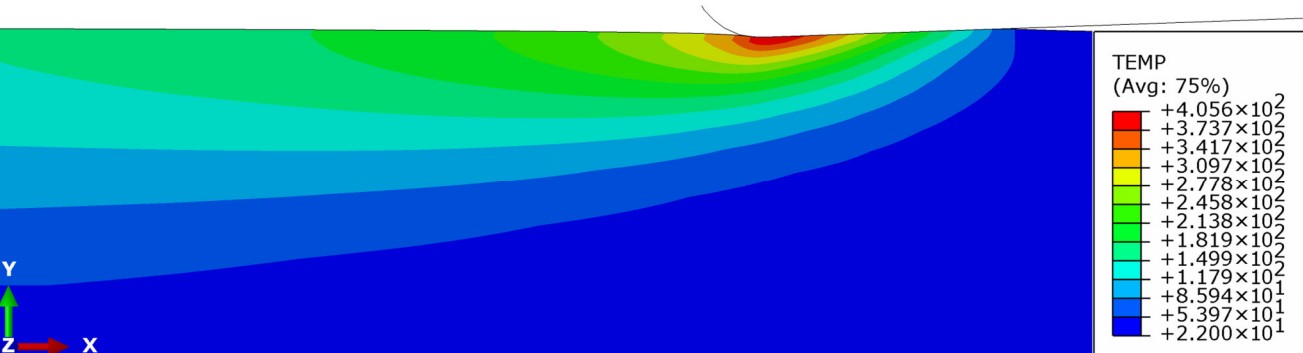

**Figure 16.** The distributions of temperature near friction contact zone (°C).

A significant increase in the values of contact pressure and cumulative deformation occurs in the case of exposure to a pre-hardened material. The deformation diagrams of such material satisfy the values of the Johnson–Cook model parameters from the article [22,23] well. In this case, the pressure in the deformation focus reaches 3.5 GPa, the degree of plastic deformation is more than 1.2, and the temperature is 405 °C, which fully corresponds to the conditions of nanostructuring of the steels given earlier.

*3.3. Investigation of the Influence of the Tilt Angle of the Cylindrical Indenter on the Microhardness, Microstructure, and Roughness of the Surface Layer*

The measurement of the microhardness $HV_{0.05}$ after nanostructuring treatment with a force of 250 N was carried out at five points of the burnishing tracks on each side of the split disc. The results of microhardness measurement depending on the indenter tilt angle are presented in Table 3 and Figure 17. The highest average value of microhardness 1508.2 $HV_{0.05}$ is reached at an angle of $\alpha = 2°$, with the lowest 1152 $HV_{0.05}$ corresponding to the tilt angle of $\alpha = 0.5$ °. An increase in the angle $\alpha$ of more than 2° leads to a decrease in microhardness. At an angle of $\alpha = 2.5°$, the microhardness is 1412.4 $HV_{0.05}$. The cause of the decrease in microhardness at an angle of $\alpha = 1.5°$ requires an in-depth additional study. The patterns of changes in microhardness in a thin surface layer generally correspond to the changes in stresses identified during finite element modeling of the process and the results of the experimental study of shear strain.

**Table 3.** Change in microhardness $HV_{0.05}$ depending on the tilt angle of the cylindrical tool $\alpha$.

| No. | Tilt Angle | $HV_{0.05}$ | | | | | Mean $HV_{0.05}$ | SE of Mean |
|---|---|---|---|---|---|---|---|---|
| 1 | 0.5 | 1152 | 1127 | 1177 | 1104 | 1203 | 1152.6 | 17.54024 |
| 2 | 1 | 1481 | 1463 | 1301 | 1379 | 1412 | 1407.2 | 32.13783 |
| 3 | 1.5 | 1362 | 1286 | 1379 | 1331 | 1257 | 1323 | 22.85388 |
| 4 | 2 | 1446 | 1463 | 1596 | 1499 | 1537 | 1508.2 | 26.95812 |
| 5 | 2.5 | 1463 | 1446 | 1379 | 1395 | 1379 | 1412.4 | 17.63973 |

The results of transmission electronic microscopy of the surface layer show that, compared with the initial state, nanostructuring burnishing with a cylindrical indenter provides a high degree of structure dispersion (Figure 18). At the same time, the indenter tilt angle has a significant effect on the degree of dispersion. Thus, at an angle of 0.5°, the grains of sizes from 40 to 85 nm are formed. The most dispersed structure is formed at the angle of 2° or 2.5°. The sizes of nanocrystals are in the range of 15 to 30 nm.

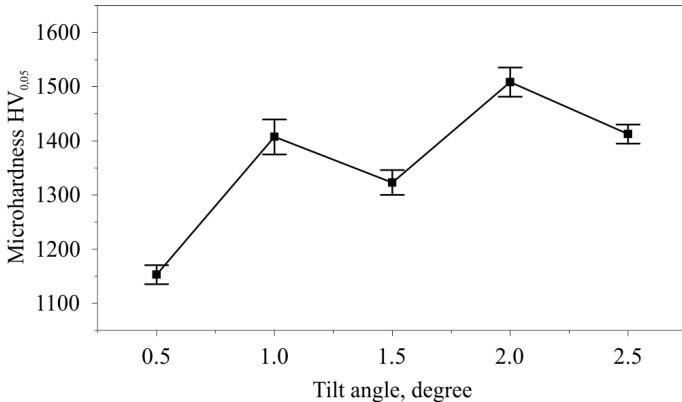

**Figure 17.** Change in microhardness $HV_{0.05}$ depending on the tilt angle of the cylindrical tool $\alpha$.

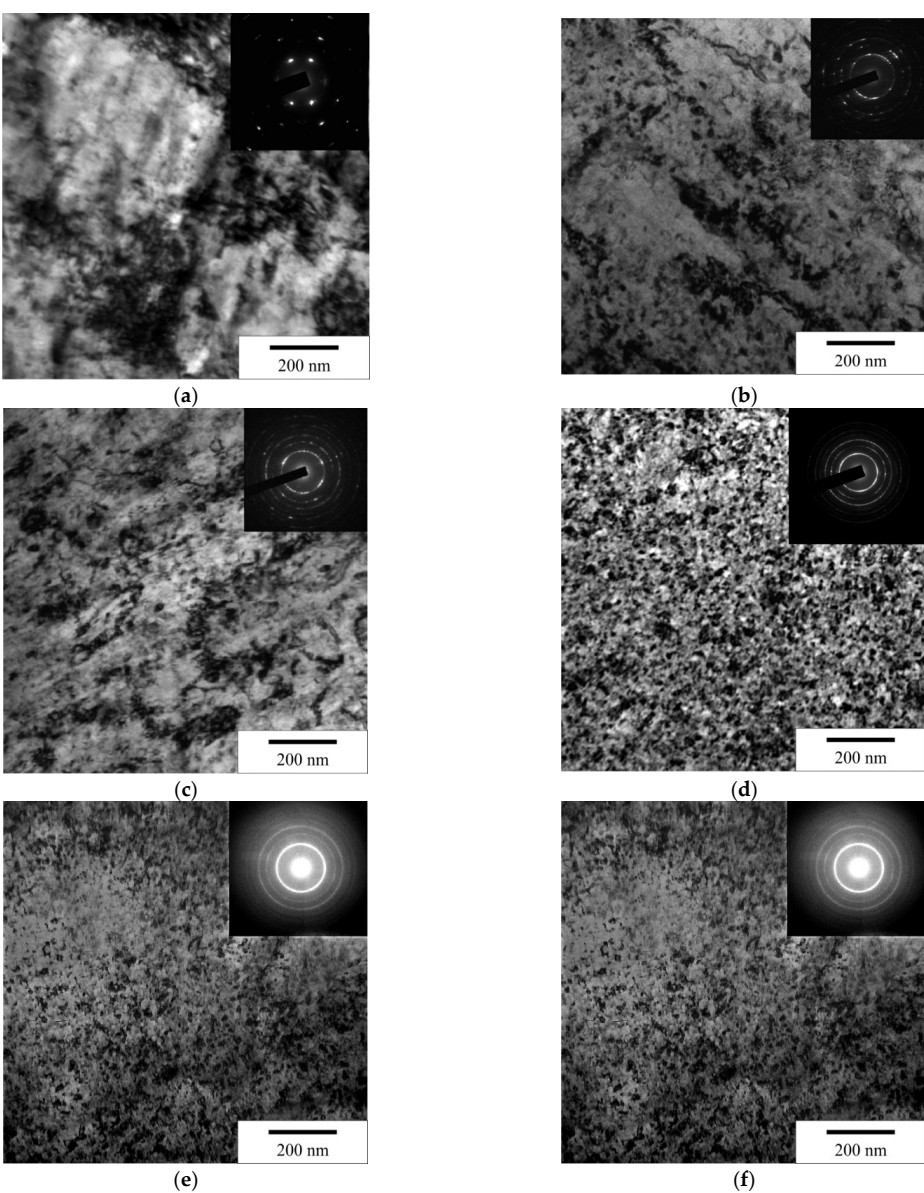

**Figure 18.** Images of the AISI 52100 steel upper layer microstructure: (**a**) the initial heat-treated state; after nanostructuring burnishing at the tilt angle of the tool: (**b**) tilt angle of 0.5°; (**c**) tilt angle of 1°; (**d**) tilt angle of 1.5°; (**e**) tilt angle of 2°; (**f**) tilt angle of 2.5°.

After burnishing with an indenter with a tilt angle of 0.5°, a minimum roughness was achieved, which was *Ra* 120 nm. With an increase in the indenter tilt angle to 1.5°, a smooth increase in roughness is observed, not exceeding *Ra* 160 nm. A further increase in the indenter tilt angle leads to a significant deterioration in roughness to the value of *Ra* 360 nm at a tilt angle of α = 2.5°. Figure 19 shows examples of surface topography at different tilt angles of the indenter. Figure 20 shows the change in the arithmetic mean deviation of the surface profile *Ra* from the tilt angle of the cylindrical indenter.

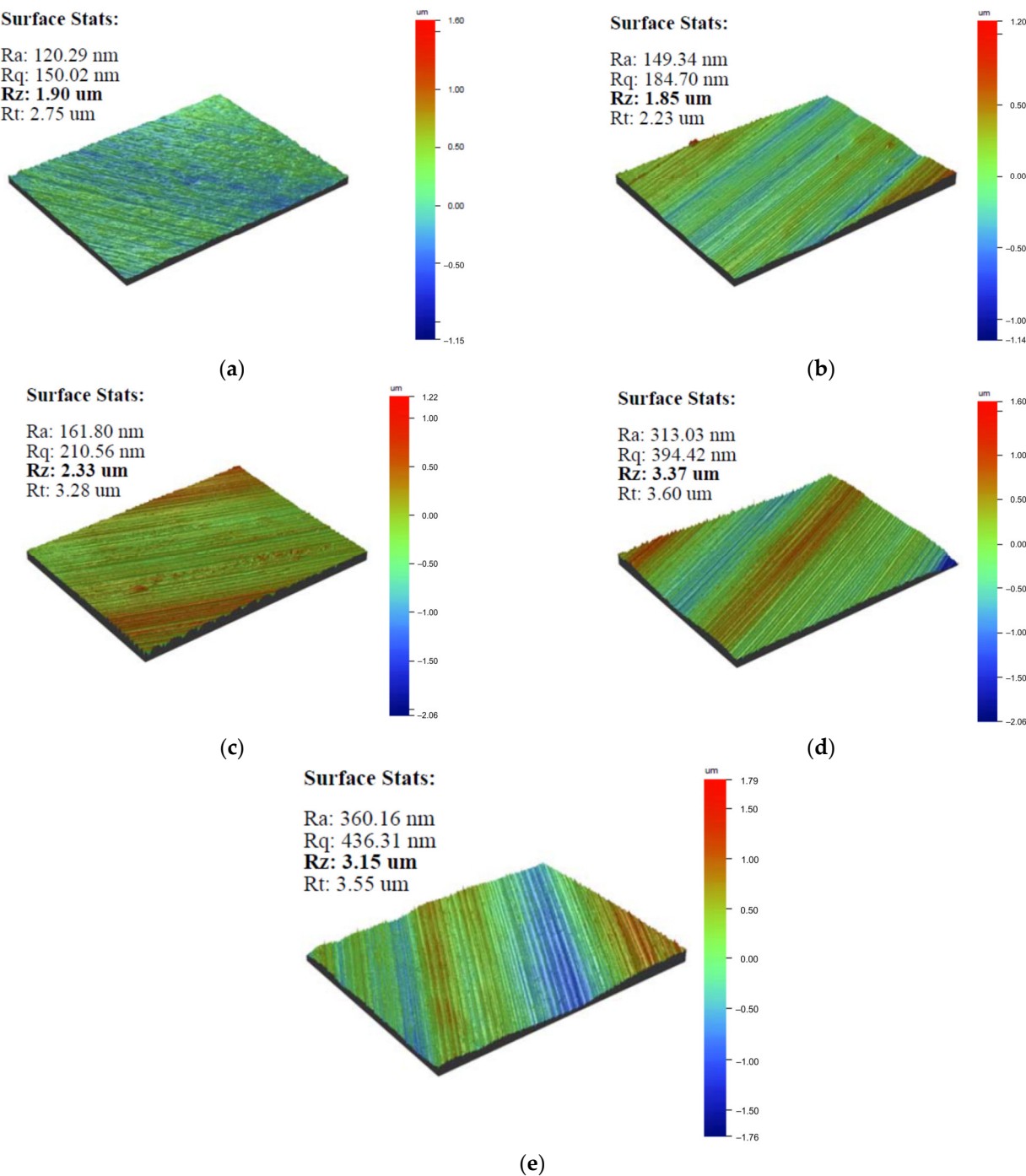

**Figure 19.** Example of a microprophile topography at different indenter tilt angles: (**a**) tilt angle of 0.5°; (**b**) tilt angle of 1°; (**c**) tilt angle of 1.5°; (**d**) tilt angle of 2°; (**e**) tilt angle of 2.5°.

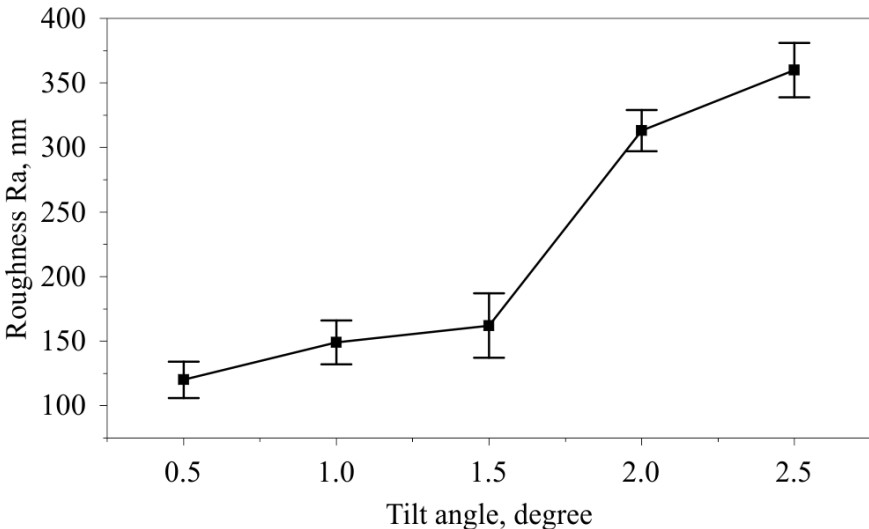

**Figure 20.** Change in the arithmetic mean deviation of the surface profile *Ra* depending on the tilt angle of the cylindrical indenter α.

## 4. Discussion

The analysis of the results of studies of dry nanostructuring burnishing of AISI 52100 hardened steel with a flat cylindrical indenter inclined at an angle of 0.5 to 2.5 degrees to the treated surface made it possible to reveal the features of change and the relationships among contact pressure, friction coefficient, and plastic deformation with the formed microstructure, microhardness, and roughness of the modified surface layer.

Modeling was performed using ABAQUS software, which is most often used for burnishing simulation. The problem was solved in a two-dimensional approximation in a dynamic coupled thermomechanical formulation, which made it possible to study in sufficient detail the stress and strain state in the contact area and the effect of elevated temperatures on it due to the formation of heat during plastic deformation and friction sliding. The presented finite element model is the development of the model proposed earlier by the authors [12]. The indenter was modeled with a rigid body, and the elastoplastic Johnson–Cook model was used to describe the behavior of steel. Significant variations in the values of the parameters of this model, which are reported in the literature, are due to the influence of steel pretreatment on its strength properties and have a great influence on the values of stresses and strains obtained in the calculations.

Finite element modeling of the process in the ABAQUS/Explicit package made it possible to establish the optimal value of the burnishing force of 250 N to provide a contact hydrostatic pressure of more than 3.5 GPa and an accumulated plastic strain $\varepsilon = 1.2$ at an indenter tilt angle of 2 degrees. In this case, the temperature in the contact zone reaches 405 °C and, according to previous studies, also contributes to the grinding of the initial structure into nanocrystallites.

A detailed study of stress distributions in the contact zone showed that during the movement of an inclined cylindrical indenter, tensile normal stresses do not form behind it, which takes place for a hemispherical indenter and could lead to cracking of the surface layers of the steel under machining. According to the literature review, the strengthening of AISI 52100 (100Gr6) quenched steel by rolling the surface with a ball with a diameter of 6 mm under a hydrostatic pressure of 400 bar gives the value 62.4 ± 0.36 HRC [29]. It is also shown that the microhardness after treatment reaches a value of 763 HV. In our case, we get a microhardness value of 1508.2 $HV_{0.05}$, which proves the advantages of the proposed approach.

## 5. Conclusions

The paper presents the results of numerical and experimental studies of the nanostructuring burnishing hardened bearing steel AISI 52100 with an innovative tool having a flat cylindrical indenter made of CBN010 from SECO, inclined at an angle to the surface being machined. Finite element simulations and experimental studies were performed under dry processing conditions with a burnishing speed of 25 m/min, while the indenter tilt angle changed from $0.5°$ to $2.5°$.

The main findings of the study are as follows:

- FEM analysis indicates that the maximum values of contact pressure and plastic strain occur when the tilt angle is $2°$. At a burnishing force of 250 N, the pressure reaches 3.5 GPa, the plastic strain is more than 1.2, and the temperature is 405 °C, which fully corresponds to the conditions of steel nanostructuring;
- Burnishing the planes of the split disk with a tool installed in a Kistler 9257BA dynamometer made it possible to establish that the friction coefficient decreases linearly from 0.33 to 0.26 as the tilt angle varies from $0.5°$ to $2.0°$;
- 3D profilometry of the mating surfaces of the split disk enabled the determination of shear strains. True strains increase from 1.05 to 2.4 within the same range of tilt angle;
- The studies at an optimal burnishing force of 250 N and a feed rate of 0.04 mm/rev reveal that the minimum value of microhardness 1,152.6 $HV_{0.05}$ corresponds to an angle of $0.5°$, while the maximum value of 1508.2 $HV_{0.05}$ occurs at a tilt angle of $2°$;
- Transmission microscopy fully explains the achieved level of microhardness since the minimum size of nanocrystallites in the range of 15–30 nm occurs at indenter tilt angles of $2–2.5°$;
- The surface roughness changes in the opposite direction to hardness, ranging from 120.3 to 161.8 nm at small tilt angles to 360.16 nm at an angle of $2.5°$.

Directions for further studies of the developed tool will be related to nanostructuring burnishing under the conditions of the cooling lubricant, which will allow the establishment of a new balance in achieving microhardness and roughness of the formed surface layer. In addition, process studies are needed for other tools and workpiece materials. It is planned to develop a 3D finite element model of the process as well.

**Author Contributions:** Conceptualization, V.K.; methodology, V.K.; software, I.S. and A.A.; validation, A.S.; formal analysis, I.S. and A.S.; investigation, V.K. and A.S.; resources, V.K. and I.S.; data curation, I.S. and A.S.; writing—original draft preparation, I.S. and A.S.; writing—review and editing, V.K.; visualization, I.S., A.S. and A.A.; supervision, V.K.; project administration, V.K.; funding acquisition, V.K. All authors have read and agreed to the published version of the manuscript.

**Funding:** The research funding from the Ministry of Science and Higher Education of the Russian Federation (Ural Federal University Program of Development within the Priority-2030 Program) is gratefully acknowledged. The numerical model was developed according to the Government research assignment for ISPMS SB RAS, project FWRW-2022-0003.

**Institutional Review Board Statement:** Not applicable.

**Informed Consent Statement:** Not applicable.

**Data Availability Statement:** The data presented in this study are available on request from the corresponding author.

**Conflicts of Interest:** The authors declare no conflict of interest.

## Appendix A

**Table A1.** Temperature dependences of elastic properties and thermal coefficient of linear expansion (TCLE) of AISI 52100 steel.

| Temperature, °C | Young's Modulus $E$, GPa | Poison's Ratio $\nu$ | Temperature, °C | TCLE $\alpha$, $10^{-6}$ Degrees$^{-1}$ |
|---|---|---|---|---|
| 22 | 201.33 | 0.277 | 22 | 11.5 |
| 200 | 178.58 | 0.269 | 204 | 12.6 |
| 400 | 162.72 | 0.255 | 398 | 13.7 |
| 600 | 103.42 | 0.342 | 704 | 14.9 |
| 800 | 86.87 | 0.396 | 804 | 15.3 |
| 1000 | 66.88 | 0.490 | | |

**Table A2.** Temperature dependences of density and thermo-mechanical properties of AISI 52100 steel.

| Temperature, °C | Density $\rho$, kg/m$^3$ | Temperature, °C | Thermal Conductivity $k$, W/(m $\times$ °C) | Temperature, °C | Specific Heat $c_p$, J/(kg $\times$ °C) |
|---|---|---|---|---|---|
| 0 | 7834 | 0 | 37.5 | 0 | 486 |
| 100 | 7809 | 100 | 40.5 | 100 | 519 |
| 200 | 7781 | 200 | 40.0 | 200 | 544 |
| 300 | 7749 | 300 | 38.0 | 300 | 578 |
| 400 | 7713 | 400 | 36.5 | 400 | 615 |
| 500 | 7675 | 500 | 34.5 | 500 | 662 |
| 600 | 7634 | 550 | 33.0 | 600 | 745 |
| 700 | 7592 | 600 | 32.0 | 700 | 2089 |
| 800 | 7565 | 650 | 30.0 | 750 | 649 |
| 900 | 7489 | 700 | 28.5 | 800 | 657 |
| 1000 | 7438 | 750 | 25.5 | 900 | 619 |
| 1100 | 7388 | 800 | 24.5 | 1000 | 636 |
| 1200 | 7340 | 850 | 25.0 | 1100 | 649 |
| 1270 | 7302 | 900 | 25.5 | 1200 | 665 |
| 1450 | 7026 | 1270 | 29.0 | 1270 | 672 |
| 1500 | 6995 | 1450 | 39.3 | 1450 | 765 |
| 1600 | 6934 | 1538 | 40.3 | 1480 | 777 |
| | | 1627 | 41.5 | 1510 | 791 |
| | | | | 1540 | 804 |
| | | | | 1600 | 804 |

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
