# Peer review of "Finite Element Simulation and Experimental Investigation of Nanostructuring Burnishing AISI 52100 Steel Using an Inclined Flat Cylindrical Tool"

_applsci, doi:10.3390/app13095324_

Round 1

Reviewer 1 Report

The paper presents the results of numerical and experimental studies of the  nanostructuring burnishing hardened bearing steel AISI 52100 with an innovative tool. Authors developed numerical and finite element simulations of the burnishing process, prepared dynamometric studies of the process and made parametric studies of the process. The article presents a suitable scientific level. However, several issues need to be corrected before publication:

- the Materials and Methods chapter should be divided differently because it contains both the assumptions of the experiment and the constitutive equations of the simulation, numerical assumptions. Separate numeric modeling from experimentation.

  - the description of the numerical model with the diagram of boundary conditions is missing,

- the test results should be divided into the results of simulation and experimental tests with the validation of FEM models.

- Has the condition of the material after pretreatment been taken into account in the models?

- Has the influence of process conditions on the location of the Belyaev point been analysed?

- Do the five material constants A, B, n, C, m in the J-C model depend on the strain rate?

- J-C model describes yield stress  (σ yield). I think it should be noted as σy

I think that the corrections increase the attractiveness of the publication.

Reviewer 2 Report

The focus of this paper is on finding optimal processing parameters in the Nanostructuring Burnishing procedure using the Finite Element Method (FEM) and experimental investigation. While the paper contains enough content, revisions are needed in the abstract, conclusions, and some numerical details from the reviewer's perspective.

(1) The authors should emphasize the purpose of the paper, which is to identify the optimal processing parameters in the burnishing process. In doing so, they should provide more background information, including the current challenges, the need for optimization, and the possible applications of such findings. The authors may modify the abstract and add more bullet points to the conclusion section to achieve this.

(2) The relation between experiments and FEM simulation is not clear. To enhance readability, the authors should consider adding a flow chart that explains the connection between the two.

(3) For the FEM simulation part, the authors may provide a mesh plot to provide clearer information on the model. Additionally, they should mention why plain strain 2D elements were chosen for the simulation.

Reviewer 3 Report

The manuscript is in decent shape only. However, the authors need to address the below queries:

1. The introduction need to be refined by including some relevant references and can be made crispy.

2. Few details are missing in materials  and methods section such as TEM etc.

3. Too long Materials and Methods section and not proportionate Results and Discussion sections.

4. too many number of images. Must combine images and reduce the total number of figures significantly.

5. Few references are not as per the format. Need to be rechecked.

Reviewer 4 Report

The paper is of sufficient scientific interest in the concerned field and the subject of the paper is related to the aim and scope of the journal. The paper is well organized. The introduction section is informative. The manuscript is clear and concise. The references are adequate, satisfactory, and given correctly. But there are some issues missing in this paper and hence, the manuscript needs minor revision based on the following comments:

1)  It is suggested to briefly start with the need of study in the abstract.

2) Flow of information has not been maintained in the literature review. The authors should have clearly indicated the objective of the present work with reference to the previous works. This would have substantially increased the importance of this research work. The authors should to explain better what are their innovative contribute?

3) Authors should clearly explain about their originality. How does the present paper differ from other related papers..

 4) I would like to know how the dimensions of the workpiece were chosen such that the effects of model boundaries on the residual stress in the interested area were negligible. I would like to see evidence of this because the target seems in fact small relative to the indenter (Fig.3) (so I would expect edge effects).

 5) I would like to see the mesh independence results of your model. The authors may benefit from the following reference.

 Dynamic Finite Element Analysis on Single Impact Plastic Deformation Behavior Induced by SMAT Process in 7075-T6 Aluminum Alloy. Met. Mater. Int. 27, 2600–2613 (2021). https://doi.org/10.1007/s12540-020-00951-y

 6)      It is recommended to include a Conclusion section showing the main findings of the study.

7)      Language needs to be refined throughout the manuscript.

Round 2

Reviewer 3 Report

The authors could address to majority of the comments: However, they need to address listed below:

1. The sample preparation methodology in TEM has not been provided.

2. There is no coherency in the introduction part. Probably minor rearrangements can make it more readable.

3. Some plots can be merged wherever the same x-axis content is there by using dual y-axes.

4. The content/legend in the distribution plots obtained from FEM are not legible.

5. Why the microhardness value at 1.5 deg tilt angle has a drop. Need detailed explanation.
